# Synthesis and characterization of novel ssDNA X-aptamers targeting Growth Hormone Releasing Hormone (GHRH)

**Burcu Ayhan-Sahin[1]◉, Zeynep-Elif Apaydın[1]◉, Pınar Obakan-Yerlikaya[2], Elif-Damla Arisan[3], Ajda Coker-Gurkan[4]\***

**1** Department of Molecular Biology and Genetics, Istanbul Kultur University, Istanbul, Turkey, **2** Department of Biomedical Engineering, Biruni University, Istanbul, Turkey, **3** Institute of Biotechnology, Gebze Technical University, Gebze, Kocaeli, Turkey, **4** Department of Molecular Biology and Genetics, Biruni University, Istanbul, Turkey

◉ These authors contributed equally to this work.
\* ajdaaaacoker@gmail.com

## Abstract

### Background

Growth Hormone Releasing Hormone (GHRH), 44 amino acids containing hypothalamic hormone, retains the biological activity by its first 29 amino acids. GHRH (NH2 1–29) peptide antagonists inhibit the growth of prostate, breast, ovarian, renal, gastric, pancreatic cancer *in vitro* and *in vivo*. Aptamers, single-strand RNA, or DNA oligonucleotides are capable of binding to target molecules with high affinity. Our aim in this study is to synthesize and select X-aptamers against both GHRH NH2 (1–29) and GHRH NH2 (1–44) and demonstrate synthesized aptamers' target binding activity as well as serum stability.

### Methods and results

Aptamers against GHRH NH2 (1–44) and NH2 (1–29) peptides were synthesized, and binding affinity ($K_d$) of 24 putative X-aptamers was determined by the dot-blot method, co-immunofluorescence staining and, SPR analysis. The serum stability of TKY.T1.08, TKY1.T1.13, TKY.T2.08, TKY.T2.09 X-aptamers was 90–120 h, respectively. The dose-dependent binding of TKY1.T1.13, TKY.T2.08, TKY.T2.09 X-aptamers on GHRHR in MIA PaCa-2 was approved by co-IF assay results. Moreover, SPR analysis indicated the $K_d$ (4.75, 1.21, and 4.0 nM) levels of TKY2.T1.13, TKY.T2.08, TKY.T2.09 putative X-aptamers, respectively.

### Conclusion

Our results illustrate the synthesis of 24 putative X-aptamers against both GHRH NH2 (1–44) and NH2 (1–29) peptides and TKY1.T1.13, TKY.T2.08, TKY.T2.09 X-aptamers have high serum stability, high target binding potential with low $K_d$ levels.

**Data Availability Statement:** All relevant data are within the paper and its Supporting information files.

**Funding:** The project was funded by Scientific and Technological Research Council of Turkey (TUBİTAK) with a project number 117Z254.

**Competing interests:** The authors have declared that no competing interests exist.

## Introduction

Cancer is a complex disease in which cells undergo malignant transformation *via* various genomic and proteomic alterations, leading to uncontrollable cellular growth and proliferation [1]. Globally, it is foreseen that one in eight men and one in ten women will die due to cancer in the 2020s [2]. Sustaining proliferation, inducing angiogenesis, and activating invasion/metastasis is essential hallmarks of cancer, according to Hanahan and Weinberg [3]. Many growth factors are involved in these processes, such as Insulin-like growth factor-I (IGF1), Epidermal Growth Factor (EGF), Transforming Growth Factor Beta (TGF-ß), Vascular Endothelial Growth Factor (VEGF) [4]. In addition, cancer cells are known to produce growth factors that induce their proliferation, and thus cell division is continually stimulated in these cells [5].

Growth Hormone Releasing Hormone (GHRH) is a hypothalamic neuropeptide that stimulates the pituitary gland for the production and secretion of Growth Hormone (GH) [6]. Although a full-size GHRH peptide comprises 44 amino acids, its biological activity resides in the first 29 amino acids in the N-terminal [7]. Instead of the neuroendocrine function, the peripheral expression of GHRH and its receptor is evident in various surgical samples of the prostate, breast, ovarian, and endometrial cancers [8]. The expression and secretion of GHRH in non-pituitary cell types imply the effect of GHRH on the regulation of cell proliferation, differentiation, and carcinogenesis [9]. GHRH peptide antagonists have been shown to trigger apoptotic cell death *via* inhibiting the GHRH signaling in the prostate, endometrial, colon, lung cancer *in vitro* and *in vivo* [6, 10]. In order to generate GHRH antagonists, we select aptamers due to their excessive specificity, high binding affinity, low toxicity, and non-immunogenic properties [11].

Aptamers are single-strand nucleic acid molecules (DNA or RNA) that can bind to target molecules such as proteins, peptides, carbohydrates, bacteria, viruses, and cancer cells for detection and diagnosis [11]. However, they are generally used as biosensors for detecting and diagnosing target molecules. VEGF aptamer (Macugen) is used for the treatment of macular degeneration [12]. The synthesis of aptamers is commonly performed by the SELEX (systemic evolution of ligands by exponential enrichment) method, a repeating amplification of nucleic acid with high binding affinity against the target molecule [13]. Nucleic acid modifications, truncations, labeling of aptamers increased the binding affinity of aptamers [11]. Recently, new generation aptamer synthesis has been performed using a magnetic bead-dependent modified ssDNA library for target binding and amplifying candidate sequences termed X-aptamer technology [14]. The most improved advantage of X-aptamer technology is synthesizing up to 5 different targets simultaneously with one SELEX method. Besides, modified nucleotides are included in the X-aptamer library to increase the stability and specificity of aptamers. However, the limitation of X-aptamer technology is that the molecular size of the target molecule or targets longer than 10 amino acid is assumed to be more preferred [14]. As there is no GHRH antagonist aptamer, we preferred to synthesize aptamers against GHRH 1–44 and 1–29. To synthesize X-aptamers that can capture every epitopic region, we selected full GHRH protein NH2 (1–44). Besides, as GHRH peptide antagonist select 1–29 region, we used GHRH NH2 (1–29) peptide as a target to synthesize X-aptamers. To illustrate, by comparing the binding affinity of GHRH antagonist X-aptamers, we aimed to synthesize, select X-aptamers against both GHRH NH2 (1–44) and NH2 (1–29) peptides.

## Materials and methods

### Cell lines and antibodies

HEK293 (CRL-1573), MIA PaCa-2 (CRL-1429), HT-29 (HTB-38), PC3 (CRL-1435), LNCaP (CRL-1740), and PNT1a (CRL-11609) cells were purchased from American Type Culture

Collection (ATCC, Manassas, VA, USA). Each cell line was grown in MEM, DMEM, McCoy's 5A, and RPMI medium (PAN Biotech, Aidenbach, Germany) completed by 10% fetal bovine serum (Gibco, Paisley, PA4, United Kingdom), 10000 U/mL penicillin, and 10 mg/mL streptomycin (PAN Biotech, Aidenbach, Germany) at 37°C in 5% $CO_2$ incubator (HERAcell 150, Thermo Scientific (Paisley, PA4, United Kingdom), respectively. Anti-His tag (1:1000), anti-GAPDH (1:1000) primary antibodies, HRP-conjugated anti-rabbit secondary antibody (1:3000), and HRP-conjugated anti-goat secondary antibody (1:3000) were obtained from Cell Signaling Technologies (Danvers, MA, USA). Anti-GHRH primary antibody (1:500) and anti-GHRHR antibody (1:1000) were purchased from Origene (Rockville, USA) and Santa Cruz Biotechnology (Dallas, Texas, USA), respectively. His-tagged (pCMV3-SP-N-His-NCV; CV023) and His-tagged GHRH (pCMV3-SP-His-ORF His-GHRH) plasmids were purchased from Sino Biological (Wayne, PA, USA).

## Bacterial His-GHRH (1–44) protein isolation and purification

$1.5 \times 10^8$ cells/mL *E. coli* HB101 become competent by the $CaCl_2$ method [15] and then 500 μg/mL His-tagged and His-tagged GHRH plasmids transformed by the heat shock method [16]. Following the transformation, His- and His-GHRH expressing selective positive controls were grown at 37°C overnight; the pellets were obtained by centrifugation and resuspended in lysis buffer (20 mM Tris-HCl pH: 7.5, 300 mM NaCl, 2M Urea, 2% Triton X-100, 1 mg/mL lysozyme and 1X EDTA-free Protease Inhibitor Cocktail). After the sonication step, the pellets were resuspended in the inclusion body solubilization solution (20 mM Tris-HCl, 0.5 M NaCl, 5 mM imidazole, and 6 M guanidine hydrochloride), and His-GHRH protein was purified by Dynabeads™ His-Tag Isolation and Pulldown magnetic beads (Invitrogen, Paisley, UK) according to the manufacturer's instructions [17].

## Expression of His-tagged GHRH peptide in HEK293 cells

HEK293 cells were seeded at 6-well plates with a seeding density of $1 \times 10^5$ cells/well. Following cell attachment overnight, cells were transfected with 1 μg/mL His tagged (pCMV3-SP-N-His-NCV) and His-GHRH (pCMV3-SP-His-ORF) plasmids and 6 μL ScreenFect A (ScreenFect, Eggenstein-Leopoldshafen) for 48 h. After incubation, cells were selected by hygromycin (Neofroxx, Einhausen, Germany) with increasing concentration for two weeks.

## RNA isolation and real-time PCR

Total RNA was extracted in HEK293 cells and putative His, His-GHRH expressing HEK293 cells by PureZol RNA isolation reagent (Bio-Rad, Hercules, California, USA). The first-strand cDNA was synthesized by the iScript cDNA synthesis kit (Bio-Rad). Primers for the GHRH gene were 5'-TATGCAGATGCCATCTTCAC-3' and 5'- TCATCCCTGGGAGTTCCTGC-3', and the 18S gene were 5'-CTACCACATCCAAGG AAGGCA-3' and 5'-TTTTTCGTCACTACCTCCCCG-3', respectively. Real-time PCR was performed in Bio-Rad CFX Connect Real-Time PCR Detection System with SYBR Green Supermix (Bio-Rad). 18S was used as a normalization control. The fold of change was calculated according to the PFAFFL method [18]. All reactions were run in triplicate.

## GHRH ELISA assay

wt, His, His-GHRH expressing HEK293 cells, MIA PaCa-2, HT-29, LNCaP, PC3, PNT1A cells were seeded at 100 mm *Petri* dishes with a seeding density of $3 \times 10^6$ cells/dish. Following overnight cell attachment, the media were concentrated *via* filtering by Amicon Ultra centrifugal filter (Merck, Massachusetts, USA). 100 μL of concentrated cellular media were used for

GHRH ELISA assay (Elabscience, E-EL-H1146) according to the manufacturer's instructions and were measured at 450 nm wavelength by a microplate reader (Multiskan, Thermo Scientific). The secreted levels of GHRH concentration were calculated according to standards. Since we do not have unlabeled aptamers to compete with labeled aptamers, we used labeled aptamers in competitive ELISA Assay instead of detection antibodies.

## Immunoblotting

Total proteins were extracted using M-PER Mammalian protein extraction reagent (Thermo Fisher Scientific). Concentrations of total proteins were quantified by Bradford assay [19]. 80 µg cellular proteins were separated on 15% SDS-PAGE and transferred to the PVDF membrane (Thermo Fisher Scientific). Membranes were blocked by 5% non-fat dried milk in TBS-T solution (1X TBS, 0.1% Tween-20) and probed with primary antibodies against GHRH and His-tag, then incubated with secondary anti-goat horseradish peroxidase-conjugated IgG, and developed with enhanced chemiluminescence solutions, images were taken by ChemiDoc Imaging System (BioRad, Hercules, California, USA).

## Immunofluorescence staining

wt, His-, His-GHRH expressing HEK293 cells, and MIA PaCa-2, HT-29, LNCaP, PC3, PNT1A cells were seeded with a seeding density of 1 x $10^5$ cells/well. Following the 48 h incubation period, cells were fixed (ice-cold methanol), permeabilized (0.1% Triton-X-100 in PBS), blocked (2% BSA in PBS) and probed with anti-rabbit anti-GHRH primary antibody (1:50), then incubated with anti-rabbit Alexa Fluor 488-conjugated Ig (1:250 dilution). Nuclear staining performed by 1 µg/mL DAPI and GHRH localization was observed fluorescence microscopy (Olympus, Japan). 1 x $10^5$ cells/well MIA PaCa-2 cells were seeded with a seeding density, incubated with selected x-aptamers (500 nM) for 72 h. Cells were fixed (ice-cold methanol), permeabilized (0.1% Triton-X-100 in PBS), and blocked (2% BSA in PBS), probed with appropriate primary antibodies (anti-mouse anti-GH primary or anti-goat anti-GHRHR primary antibody), then incubated with anti-mouse Alexa Fluor 555-conjugated secondary antibody or FITC-conjugated anti-goat secondary antibody, respectively. Cellular GH or GHRHR expressions were visualized by fluorescence microscopy (Olympus, Japan).

## X-aptamer selection and synthesis

According to the instruction, X-aptamer selection was performed by the X-aptamer selection kit (AM Biotechnologies, Houston, USA). At the first step, the X-aptamer library was prepared as follows: the library resuspended in a selection buffer [1X PBS pH:7.4, 1 mM MgCl$_2$, 0.05% Tween20, 0.02% BSA], incubated at 95°C water bath for 5 minutes, and cooled down slowly to room temperature. We performed two a bead based aptamer selection; first a bead based aptamer selection, we used bacterial/eukaryotic GHRH 1–44 His-tagged peptides, and second a bead based aptamer selection used GHRH 1–29 peptide (Sigma, G6771) as a target. Due to kit instructions, for the first a bead based aptamer selection, 10 µg His-tagged bacterial and 10 µg eukaryotic His-tagged GHRH 1–44 peptide was used to pulldown by magnetic beads. In addition, for the second bead based aptamer selection, biotinylated GHRH 1–29 peptide was used for streptavidin magnetic bead isolation. Next, His-tag/biotinylated isolation and pulldown magnetic beads were prepared for negative selection by washing with a selection buffer and incubated with the prepared library for one hour at room temperature. After incubation, the library which did not bind to magnetic beads (i.e., negatively selected library) was transferred to a clean tube with the aid of a magnetic stand. Then, his-tagged target protein [GHRH 1–44 (bacterial and eukaryotic)] or biotinylated GHRH 1–29 was mixed with His-tag/streptavidin

coated-magnetic beads, incubated for 30 minutes at room temperature, and target protein-bound magnetic beads were washed with a selection buffer. Next, negatively selected library and target-bound magnetic beads were incubated at room temperature for 90 minutes. Magnetic beads were washed with a selection buffer and resuspended in the selection buffer. Putative aptamers were cleaved by incubating with NaOH at 65˚C for 30 minutes and neutralizing with Tris-HCl. Cleaved putative aptamers were transferred to a clean tube. Spin column buffer exchange was performed to eliminate NaOH, and a "cleaved oligonucleotide pool" was obtained. For the secondary selection process, four 1.5 mL centrifuge tubes were labeled Tube 1, Tube 2, Tube 3, Tube 7. Cleaved oligonucleotide pools and selection buffers were added to all four tubes. His-GHRH (1–44) protein was added to Tube 2 and Tube 3 and incubated with rotation at room temperature for one hour. His magnetic beads were added to Tube 2, Tube 3, and Tube 7 and incubated with rotation at room temperature for 30 minutes. 10 μL from the tubes used as a template for PCR reaction. The forward primer was common for all; however for tube 1, reverse primer 1, tube 2 reverse primer 2, tube 3 reverse primer 3, and tube 7 reverse primer 7 was used. PCR reaction prepared in total 100 μL volume as follows: 1X PCR Buffer, 2.5 mM $MgCl_2$, 0.2 mM dNTP, 0.4 μM forward primer, 0.4 μM indicated reverse primer, 1U Taq polymerase. The temperature condition was 94˚C 1 minute, cycles of 94˚C 30 seconds, 50˚C 30 seconds, 72˚C 1 minute, and the last extension at 72˚C for 3 minutes. For each tube, 14, 18, 22 cycles of PCR were performed. PCR products were run on agarose gel, and images were obtained by UV Imaging System. For the four tubes, 25 cycles were performed, and all PCR products were mixed in one tube and resent back to the manufacturer for Illumina next-generation sequencing (AM Biotechnology, Texas, Huston, USA). Next-generation sequencing and synthesis of candidate x-aptamers with 5' biotin tag were performed by the corresponding company (AM Biotechnology, Texas, Huston, USA) [14].

## Dot-blot assay

Putative 24 X-aptamers were renaturated (95˚C for 10 min), incubated with either GHRH NH2 (1–29) or GHRH NH2 (1–44) peptide in dot blot binding buffer (15 mM Tris, 100 mM NaCl, 1 mM $MgCl_2$) overnight at +4˚C, blotted to nitrocellulose membrane (Invitrogen) as a small spot. As a negative control we used scramble aptamer which has a ΔG: -27.621 kJ (5′ – TTTTTTTCAGACCAGCCGTGCACGACGAACCACAAGCAGGTGGGCCCA–3′) Following air dry of the membrane, HRP-Streptavidin conjugate (Thermo Fisher Scientific) was added onto each membrane spot, incubated for two hours at room temperature in dark, TMB substrate solution (Thermo Fisher Scientific) added, and images were taken by LCD Camera. The same protocol was performed for $K_d$ determination of each putative aptamers by the dot-blot method. The $K_d$ levels were determined in each aptamer dot spot intensity level by Sigma Plot v14.0 (https://systatsoftware.com/products/sigmaplot/) program.

## Serum stability of putative X-aptamers

5 μL from each 100 μM putative X-aptamer against GHRH was mixed with 5 μL human serum (Sigma Aldrich, P2918, Massachusetts, USA) for 0–120 h at 37˚C. Following each incubation period, 10 μL of the mixture loaded at 12% polyacrylamide gel electrophoresis and was visualized under the ChemiDoc MP imaging system (Bio-Rad Laboratories, Hercules, USA). DNaseI digested aptamers were used as positive controls.

## Cyclic AMP assay analysis

To determine the impact of aptamers on cAMP levels, we performed Cyclic AMP XP® Assay Kit (CST, 4339). Both HT29 colon cancer and MIA Paca-2 pancreatic cancer cells were seeded

at 12 well-plate with a density of $2 \times 10^4$. Following cell attachment, adherent cells treated with selected X-aptamers for 72 h, and cAMP assay performed according to the manufacturer's instructions.

## GH and GHRHR expression profile by immunofluorescence

$1 \times 10^5$ cells/coverslip of MIA Paca-2 cells were seeded on coverslips at 6 wells. After cell attachment, 500 nM selected X-aptamers were treated. Following fixation, permeabilization, blocking and probing anti-goat anti-GH primary antibody (1:50 dilution) at +4˚C overnight, anti-goat Alexa flour 555-conjugated secondary antibody (1:250) applied at +4˚C for 1 h. Following washing, blocking, cells probed with anti-mouse anti-GHRHR primary antibody (1:50 dilution) at +4˚C overnight. Then, anti-mouse FITC conjugated secondary antibody (1:250 dilution) was applied at +4˚C for one hour. The nucleus was stained with DAPI and images visualized by fluorescence microscopy (Olympus, Japan).

## Dose-dependent X-aptamer binding affinity by immunofluorescence

MIA PaCa-2 pancreatic cancer cells were seeded on coverslips with a density of $1 \times 10^5$ cells/ coverslip. Following cell attachment, x-aptamers were applied in a dose-dependent manner (0–500 nM), cells were fixed, permeabilized, blocked, and probed with anti-rabbit anti-GHRH primary antibody (1:50 dilution) at +4˚C overnight. Then, the anti-rabbit Alexa Fluor 568-conjugated secondary antibody (1:250 dilution) was applied at +4˚C for one hour, 1 μL streptavidin solution added, and incubated at 37˚C for 30 min, and probed with Streptavidin-Alexa fluor 488 conjugate (1:1000 dilution) for one hour at RT. The nucleus was stained with DAPI and images were visualized by fluorescence microscopy (Olympus, Japan). GHRH siRNA silencing with/without aptamer treatment was used as a negative control for aptamer binding affinity. 50 nM GHRH siRNA (Ambion, AM16708) with 6 μL FuGENE 6 (Promega, E2691) transfection reagent for 48 h performed in MIA Paca-2 cells. Following siRNA incubation period 500 nM aptamer treated and aptamers probed with Streptavidin-Alexa fluor 488 conjugate and GHRH probed with anti-rabbit Alexa Fluor 568-conjugated secondary antibody.

## SPR analysis

The CM-5 chip was loaded with the dose-dependent (250 nM-4000 nM) GHRH NH2 1–29 peptide (Sigma, G6771) in acetic acid solution. Following the incubation period, every three selected aptamers (TKY2.T1.13, TKY.T2.08, and TKY.T2.09 X-aptamers) with three repeats were run in BiaCore T200 (GE Healthcare Life Sciences, Uppsala, Sweden) for 120–12000 sec. Single Cycle Kinetics parameters were analyzed, and modeling was performed by kinetic analyses-heterogeneous ligands. SPR analysis was performed in SUNUM Center at SABANCI University (Istanbul-TURKEY).

## Statistical analysis

All the experiments were analyzed statistically by two-way ANOVA using GraphPad Prism 8 (GraphPad Software, La Jolla, CA, USA). p-values considered statistically significant were given as $^*p < 0.05$, $^{**} p < 0.01$, $^{***} p < 0.001$, respectively. Error bars in the graphs generated using ± standard deviation (SD) values. The mean ± SD of the data representing ELISA assay and the qRT-PCR analysis was achieved from at least two experiments with three replicates.

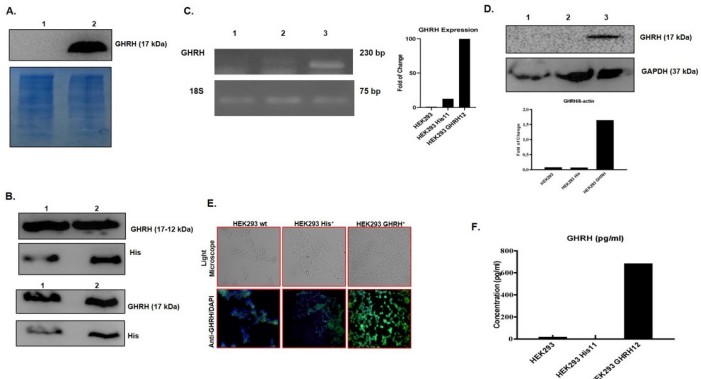

**Fig 1. Obtaining and purification of His-tagged GHRH (1–44) ligand for the aptamer selection. A** GHRH (1–44) expression of bacterial colonies expressing His- and His-GHRH vectors demonstrated by immunoblotting. **1** *E. coli* HB101 His vector clone, **2** *E. coli* HB101 His-GHRH vector clone. **B** Purified His tagged bacterial GHRH 1–44 ligand was demonstrated by immunoblotting *via* anti-GHRH and anti-His tag antibodies, respectively. **1** His-GHRH expressing total bacterial lysate, **2** His-GHRH expressing solubilized pellets. Eukaryotic GHRH NH2 (1–44) expression in His-GHRH vector-transfected HEK293 cells was identified by **C** qRT-PCR, **D** immunoblotting, and **E** immunofluorescence. **1** HEK293 wt **2** HEK293 His-transfected **3** HEK293 His-GHRH transfected. **F** GHRH 1–44 concentration in the media of HEK293 cells was measured by Human GHRH ELISA. GAPDH was used as a loading control. 18S was used as an internal control. DAPI was used to indicate the nuclei of the cells.

## Results

### Purification of His-tagged GHRH peptide

To obtain a target for X-aptamer selection, both bacterial and eukaryotic His-GHRH NH2 (1–44) expressing cellular models were generated. To yield a high range of GHRH 1–44 peptide, we selected *E.coli* HB101 cells and for post-transcriptional modifications of GHRH 1–44, we used HEK293 cells. *E.coli* HB101 cells expressing His-tagged and His-tagged GHRH protein determined by immunoblotting following plasmids transformed by heat-shock method (Fig 1a). Solubilized inclusion bodies and His-magnetic bead pull-down extracts containing His-GHRH determined by immunoblotting, respectively (Fig 1b). Since the bacterial protein synthesis mechanism lacks post-translational modifications, HEK293 cells were used for target protein production. For this purpose, His tagged (pCMV3-SP-N-His-NCV) and His-tagged GHRH (pCMV3-SP-His-ORF) plasmids were transfected to HEK293 human embryonic kidney cells, and stable His-tagged and His-GHRH-tagged expressing HEK293 cells generated following increased concentration of hygromycin application for selection. Transcriptional and translational GHRH 1–44 expression was determined by RT-PCR (Fig 1c), and immunoblotting (Fig 1d). Besides, the immunofluorescence method was confirmed the intracellular expression of GHRH majorly in HEK293-His-tagged GHRH stable cell line (Fig 1e). Moreover, the HEK293-His-GHRH stable cell line secreted 685,38 pg/mL GHRH was determined by the ELISA method (Fig 1f).

### Synthesis and selection of x-aptamers against GHRH peptide

To synthesize and select X-aptamers against both GHRH NH2 (1–44) and NH2 (1–29), we performed two a bead based aptamer selection according to the kit protocol. First bead based aptamer selection, purified 10 µg bacterial and 10 µg eukaryotic His-tagged GHRH used, cycle-course PCR performed as 14 cycles, 18 cycles, 22 cycles, and 25 cycles of last PCR performed to obtain samples for sequencing (Fig 2a). As bacterial and eukaryotic His-tagged GHRH protein has 10 Histidine amino acids at N terminal region, GHRH 1–29 peptide also

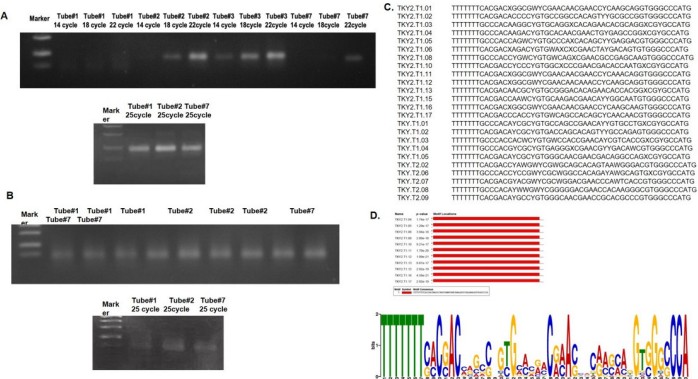

**Fig 2. Synthesis, selection of putative x-aptamers against Growth Hormone Releasing Hormone. A** First bead based aptamer selection against GHRH NH2 (1–44) target and **B** second bead based aptamer selection against GHRH NH2 (1–29) target was performed by using the X-aptamer kit. Tube #1: Cleaved oligunucleotide pool, Tube #2: Cleaved oligonucleotide pool+Magnetic beads+Prokaryotic GHRH (1–44) target, Tube #3: Cleaved oligonucleotide pool+Magnetic beads+Eukaryotic GHRH (1–44) target, Tube #7: Cleaved oligonucleotide pool+Magnetic beads. Cycle course PCR (up) and last PCR (down) amplification of the x-aptamer selection process were given. **C** Sequences of putative x-aptamers were determined by next-generation sequencing. **D.** Analysis for all 24 putative aptamers were performed by using MEME program (https://meme-suite.org/meme/).

used as a target for second bead based aptamer selection cycle-course and last PCR was also performed for GHRH (1–29) target (Fig 2b). All the samples were sequenced by Illumina next-generation sequencing *via* aligning 3' conserved sequences and putative x-aptamers were synthesized with a 5'-biotin tag by a commercial company, AM Biotechnologies (Houston, USA). The sequences of putative x-aptamers were given in Fig 2c. Aptamers against bacterial His-tagged GHRH were TKY2.T1.01, TKY2.T1.02, TKY2.T1.03, TKY2.T1.04, TKY2.T1.05, TKY2.T1.06, TKY2.T1.08, TKY2.T1.10, TKY2.T1.11, TKY2.T1.12, TKY2.T1.13, TKY2.T1.15, TKY2.T1.16, TKY2.T1.17. TKY.T1.01, TKY.T1.02, TKY.T1.03, TKY.T1.04, TKY.T1.05 aptamers were against eukaryotic GHRH NH2 (1–44) peptide. Putative X-aptamers against GHRH 1–29 target named as TKY.T2.02, TKY.T2.06, TKY.T2.07, TKY.T2.08, TKY.T2.09.

## Screening the binding affinity of putative X-aptamers to the target protein by dot blot assay and SPR analysis and illustrating the serum stability of each X-aptamer

To observe the binding affinity of X-aptamers to the target protein, the dot blot assay was performed. The dot intensities measured and fold of change of band intensities calculated by Graphpad Prism 8 program *via* using spot intensity. According to calculations, putative x-aptamers against GHRH 1–44 target TKY2.T1.01, TKY2.T1.02, TKY2.T1.03, TKY2.T1.04, TKY2.T1.05, TKY2.T1.08, TKY2.T1.10, TKY2.T1.12, TKY2.T1.13, TKY2.T1.15, TKY2.T1.17, TKY.T1.01, TKY.T1.02 and putative X-aptamers against GHRH 1–29 target TKY.T2.02, TKY.T2.06, TKY.T2.07, TKY.T2.08, TKY.T2.09 showed statistically significance in binding affinity (Fig 3a and 3b). Then binding affinity of X-aptamers to target assessed in increasing aptamer doses. The fold of change was calculated according to the dot intensities and dissociation constant ($K_d$) was calculated by nonlinear regression in GraphPad Prism software (https://www.graphpad.com/scientific-software/prism/) (Fig 3c). TKY2.T1.08, TKY2.T1.13, TKY.T2.08, and TKY.T2.09 were selected for further experiments according to obtained data. To determine the non-specific binding of selected X-aptamers against its target, we performed GHRH competitive ELISA assay and dot-blot analysis (Fig 3d). According to GHRH ELISA results, instead

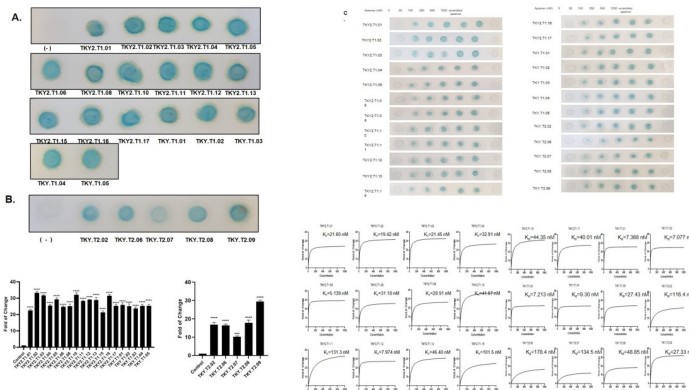

**Fig 3. Determination of binding affinity, dissociation constant (K$_d$) of putative x-aptamers.** The binding affinity of putative X-aptamers was determined by dot blot assay against **A** GHRH NH2 (1–44) target and **B** GHRH NH2 (1–29) target. **C** Dose-dependent dot blot assay was performed in increasing doses (0–1000 nM) of X-aptamers. The scramble aptamer was used a negative control for dot-blot analysis. Dot intensities were measured by Image J (imagej.nih.gov/ij/) and analyzed by GraphPad Prism 8.0. Nonlinear regression analysis was performed by Sigma Plot v14.0 and K$_d$ was calculated. **D.** The specific binding affinity of selected aptamers against GHRH peptide was determined by both GHRH sandwich ELISA assay (left panel) and dot-blot analysis (right panel) in a dose-dependent manner.

of using detection antibody, we used x-aptamers to competitively capture GHRH peptide. Dose-dependent X-aptamer exposure increases the absorbance of GHRH ELISA results in each aptamer. To confirm the ELISA results, we performed a dot-blot analysis in a dose-dependent manner. As we do not have any unlabeled aptamers, the GHRH antibody is used as a competitor for GHRH X-aptamers against GHRH peptide binding. Thus, we performed a GHRH ELISA assay. X-aptamers specifically bind to GHRH peptide like GHRH antibody and this effect sharply increased in a dose-dependent manner by both GHRH ELISA assay and dot-blot analysis (Fig 3d). To evaluate the serum stability of each putative aptamer, 0–120 h incubation of each aptamer with human serum at 37˚C, the mixture visualized by polyacrylamide gel electrophoresis. According to the serum stability results, TKY2.T1.02, TKY2.T1.04, TKY2.T1.05, TKY2.T1.08, TKY2.T1.13, TKY2.T1.17, TKY.T2.05, TKY.T1.05, TKY.T2.02, TKY.T2.07, TKY.T2.08 and TKY.T2.09 X-aptamers were stable in human serum within 120 h period at 37˚C (Fig 4a and 4b). To confirm the K$_d$ levels for selected TKY2.T1.13, TKY.T2.08 and TKY.T2.09 X-aptamers, we performed SPR analysis. According to SPR analysis, K$_d$ levels for TKY2.T1.13, TKY.T2.08, and TKY.T2.09 were determined as 4.75, 1.21, and 4.0 nM, respectively (Fig 4c).

## GHRH expression and secretion profile of MIA PaCa-2, HT-29, PC3 cancer cells and PNT1A epithelial derived normal cell

To determine the binding affinity of X-aptamers on cellular GHRH target, at first, GHRH-expressing cells were determined. Pancreatic cancer cell line MIA PaCa-2, colorectal cancer cell line HT-29, prostate cancer cell line PC3 and LNCaP, and normal human prostate epithelial cell line PNT1a used for this purpose. Primarily, GHRH expression in these cell lines determined by immunofluorescence assay with anti-GHRH antibody and it was observed that all these cell lines expressed GHRH endogenously (Fig 5a). GHRH expression of these cells was also investigated by immunoblotting (Fig 5b). GHRH concentration in media was also calculated by human GHRH ELISA assay. GHRH concentration in media was about 1500 pg/mL in MIA PaCa-2, HT-29, and PC3 cells while the value is about 700 pg/mL in PNT1a cells (Fig 5c).

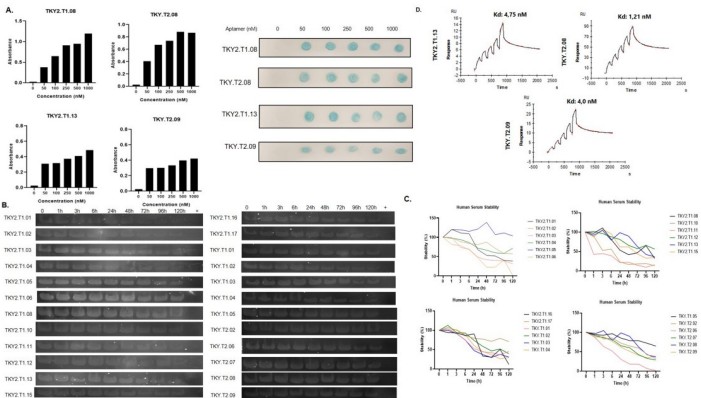

**Fig 4. Determination of dissociation constant (K_d), and serum stability of putative x-aptamers. A.** Stability of each putative aptamer detected by polyacrylamide gel electrophoresis of serum and aptamer mixture at 37˚C in time-dependent (0–120 h) manner. Each experiment was performed and repeated at least three times, given figures are the representative figure of one of the three assay repeat results. ns nonspecific, *p<0.05, **p<0.01, ***p<0.001, ****p<0.0001 **B.** Binding affinity of selected x-aptamers to their specific target was demonstrated by Surface Plasmon Resonance analysis. After loading the CM-5 chip with the dose-dependent target molecule and incubation period, each aptamer was run in BiaCore T200 in triplicate for 120–12000 seconds.

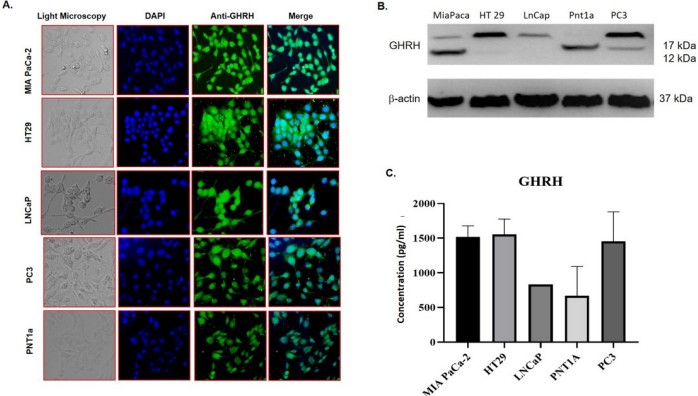

**Fig 5. Demonstration of GHRH expression in different cancer cell lines. A** GHRH expression was figured out in MIA PaCa-2 pancreatic cancer, HT-29 colorectal cancer, PC3 prostate cancer, and PNT1a normal prostate epithelium cell lines by immunofluorescence. DAPI was used to observe the nuclei of the cells. **B** Translational expression of GHRH in MIA PaCa-2, HT-29, PC3, and PNT1a cells was determined by immunoblotting. ß-actin was used as a loading control. **C.** GHRH concentrations in media of MIA PaCa-2, HT-29, PC3, and PNT1a cells determined by Human GHRH ELISA.

## Determination of dose-dependent binding affinity of x-aptamers on MIA PaCa-2 cells by immunofluorescence staining

To screen the dose-dependent binding affinity of TKY2.T1.08, TKY2.T1.13, TKY.T2.08, and TKY.T2.09 X-aptamers on MIA PaCa-2 cells, immunofluorescence was performed by using streptavidin-Alexa Fluor-488 as selected aptamers labeled with biotin at 5'-end. Due to immunofluorescence staining, both TKY2.T1.08 and TKY.T2.08 x-aptamers binding on MIA PaCa-2 membrane determined only after 50 nM X-aptamer treatment, and binding affinity increased in a dose-dependent manner [Fig 6a and 6e (left panel)]. However, significant binding was observed after 250 nM TKY2.T1.13, and 100 nM TKY.T2.09 X-aptamer treatments

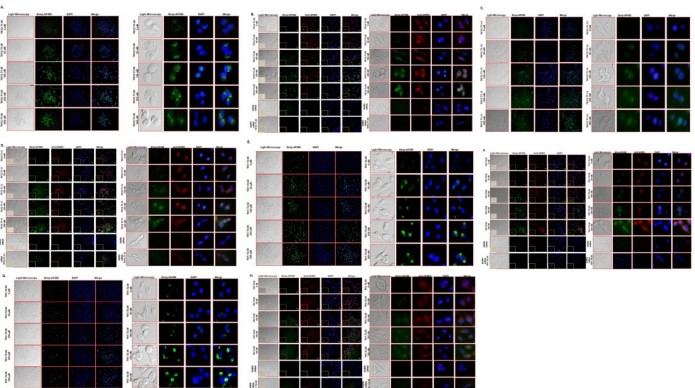

**Fig 6. Investigation of the binding location of X-aptamers in MIA PaCa-2 cells.** The binding position of **A** TKY2.
T1.08, **C** TKY2.T1.13, **E** TKY.T2.08, **G** TKY.T2.09 x-aptamers were investigated by immunofluorescence. Streptavidin-
Alexa Fluor 488 conjugate was used to specify biotin-labelled x-aptamers. Cells were treated with X-aptamers in a
dose-dependent manner (0–500 nM). Following incubation, cells were examined under a fluorescence microscope
(Olympus) and images were taken and analyzed. The binding position of **B** TKY2.T1.08, **D** TKY2.T1.13, **F** TKY.T2.08,
**H** TKY.T2.09 x-aptamers investigated by co-immunofluorescence. Streptavidin-Alexa Fluor 488 conjugate was used to
specify biotin-labelled X-aptamers. Streptavidin-Alexa Fluor 588 conjugated anti-GHRH was used to specify GHRH.
Cells were treated with x-aptamers in a dose-dependent manner (0–500 nM). Following incubation, cells were
examined under a fluorescence microscope (Olympus) and images were taken and analyzed. GHRH siRNA was used
as a negative control.

(Fig 6c and 6g). To demonstrate the binding of X-aptamers with GHRH ligand, we performed
co-immunofluorescence staining with both Streptavidin-488 (aptamer detection) and Alexa-
Fluor 565 (GHRH detection) [Fig 6b, 6d, 6f and 6h (right panel)]. Similar to screening immu-
nofluorescence staining, each aptamer binding was determined on MIA PaCa-2 cells simulta-
neously with GHRH co-staining. Moreover, we demonstrated that silencing of GHRH
expression inhibited X-aptamer binding by immunofluorescence assay (Fig 6b, 6d, 6f and 6h).

## Blocking activity of GHRH signaling of selected aptamers

To evaluate the selected aptamers' effect on GHRH signaling, we performed cAMP assay anal-
ysis and GH/GHRHR co-immunofluorescence staining (Fig 7a and 7b). Due to cAMP assay
analysis, both TKY2.T1.08 and TKY2.T1.13 X-aptamers significantly depleted intracellular
cAMP concentration as compared to untreated control cells in HT-29 cells. When we checked
the results, no significant effect was observed for cAMP levels for TKY2.T1.08 and TKY2.
T1.13 X-aptamers application in MIA Paca-2 cells. In addition, we measured a statistically sig-
nificant decline in cAMP levels following TKY.T2.08 and TKY.T2.09 X-aptamers treatment in
MIA Paca-2 cells (Fig 7a). To demonstrate the blocking activity of selected aptamers on
GHRH signaling, we performed GH/GHRHR co-immunofluorescence staining (Fig 7b).
Active GHRH signaling due to GH and GHRHR expression profile was detected in untreated
and GHRH peptide treated MIA Paca-2 cells. However, a significant downregulation on GH/
GHRHR expression demonstrated following TKY2.T1.08, TKY.T2.08, and TKY.T2.09 treat-
ment in MIA Paca-2 cells (Fig 7b).

## Discussion

GHRH is a neuropeptide that is known to be ectopically expressed in many cancer types such
as the pancreas [20], prostate [21], ovarian [22] and lung cancer [23]. Due to the expression of
GHRH/GHRHR and biologically active signaling leads the potential essential role of

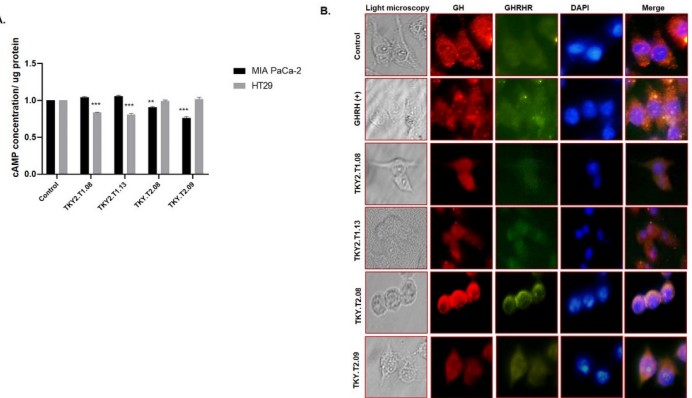

**Fig 7. Blockage of GHRH signaling by selected aptamers. A.** The effect of TKY2.T1.08, TKY2.T1.13, TKY.T2.08, TKY.T2.09 x-aptamers on the intracellular cAMP levels were determined by cAMP assay kit in both HT29 and MIA Paca-2 cells. **B.** The impact of TKY2.T1.08, TKY2.T1.13, TKY.T2.08, TKY.T2.09 x-aptamer on GH/GHRHR expression profile was investigated by co-immunofluorescence assay under a fluorescence microscope (Olympus). GHRH peptide (500 nM) was used as a positive control. Each experiment was performed and repeated at least two times, given figures are the representative figure of one of the three replicated assays. ns: non-specific, *p<0.05, **p<0.01, ***p<0.001, ****p<0.0001.

therapeutic regimes targeting GHRH. For this purpose, many GHRH peptide antagonists have been designed and synthesized by modifications on GHRH NH2 (1–29) peptide [24]. It was shown that these antagonists compete with GHRH for binding to the GHRH receptor and block receptor activation [25]. In this way, these antagonists were demonstrated to inhibit proliferation and induce apoptosis, *via* a direct effect on tumor cells in many cancer types such as lung, breast, prostate, ovarian, colorectal, and pancreas cancers [6, 10]. With a similar strategy, we tried to target GHRH by synthesizing capturing aptamer as a ligand to block the GHRH signaling. Although GHRH antagonists were studied to block GHRH signaling, the peptide nature of these antagonists limits their use *in vivo*. Also, there are no reported aptamers against GHRH in literature, thus in this study, we aimed to select an aptamer against GHRH NH2 (1–44) and NH2 (1–29) peptide as a GHRH antagonist and characterize its binding affinities to target.

Aptamers can be easily degraded by plasma nucleases because of their oligonucleotide structure. This challenge may be overcome by some modifications to different positions of nucleotides, phosphodiester bonds, or oligomerization by using L-conformation nucleotides which are not recognized by endonucleases [11]. Aptamers are selected conventionally by a method called SELEX (systemic evolution of ligands by exponential expansion) [26, 27]. Nucleic acid aptamers are single-stranded synthetic DNA or RNA oligonucleotides that may fold into three-dimensional structures and interact with many types of targets with a high affinity and specificity [11, 13]. Many molecules including small molecules, proteins, peptides, toxins, and even cells can be targets of aptamers [28]. Although generally aptamers were preferred as a capturing key molecule in a biosensor and conjugated with chemotherapeutic agents, aptamer-based hormone targeting technology approaches still maintained. As aptamers are smaller, less immunogenic, with low cost, more stable, more prone to chemical modifications, and renaturation capability after non-physiological conditions, they are more preferable to peptides or antibodies [29]. Due to their structural advantages, aptamers against some target molecules were preferred for metabolic disease treatment. Among these aptamers, Macugen is the first FDA-approved VEGF targeting RNA-aptamer for age-related macular degeneration treatment [12].

Aptamers selected by the SELEX method might be assumed as a time-consuming method due to having a limited number of modifications, and also performing 8–15 selection rounds [29]. Therefore, new methods providing more functional groups and more stability have been developed. One of the newly-developed technologies is X-aptamers, which contain protein-like or drug-like side chains providing both more stability against nucleases and more affinity to target molecules [14]. Although X-aptamer technology and selection is a newly-emerged system, it is being used in different studies. Recently, potential biomarkers have been identified for schizophrenia by using X-aptamer technology [30]. X-aptamers against two immune checkpoint proteins PD-1 and PD-L1 have also been selected by this technology [31]. In this study, we selected x-aptamers against the growth hormone-releasing hormone by using an X-aptamer selection kit.

In the case of GHRH, it was indicated that, N-terminal 29 amino acids of the 44 amino acids-long GHRH peptide show full biological activity [7]. To study the difference, if any, we used both GHRH $NH_2$ (1–44) and GHRH NH2 (1–29) peptides for aptamer selection. Further, for purification of the target protein, we needed large-scale protein synthesis in the cells and we used *E. coli* expressing His-tagged GHRH protein for large-scale protein isolation. Also, since the bacterial protein synthesis system lacks post-translational modifications, we transfected human embryonic kidney HEK293 cells with His-GHRH vector stably and used the protein isolated from these cells as a target. Moreover, increasing technology on nucleic acid-based drug discovery, new generation aptamers synthesis by magnetic bead loaded DNA library was designed. The trademark name for this biotechnological approach was X-aptamer. Thus, we synthesized X-aptamers *via* targeting both GHRH NH2 (1–44) and NH2 (1–29) by using an X-Aptamer selection kit. Following cloning and sequencing of magnetic bead-based PCR products, 24 putative X-aptamers were synthesized by AM BioTechnologies (Houston, USA) (Fig 2c). Mfold program is generally preferred for the determination of ssDNA Aptamer's schematic profiles and ΔG levels under physiological conditions. As we used new generation technology for synthesizing aptamers; X-aptamer, they are containing modified nucleotides such as phosphorodithioate, indole- or phenol-modified deoxyuridine. So there is not any tool that illustrates the potential secondary structure predictions of X-aptamers. To determine the binding affinity of aptamers against its synthesized ligands, Sypabekova et. al, 2017 and Li et. al, 2017 preferred dot-blot methods [32, 33]. As our 24 putative X-aptamers have 5' biotinylated labeling, we performed the dot-blot method *via* targeting GHRH NH2 (1–44) and GHRH NH2 (1–29) peptide (Fig 3a and 3b). According to the dot-blot spot intensity relative folding with aptamer absent spot intensity, TKY2.T1.01, TKY2.T1.02, TKY2.T1.03, TKY2.T1.04, TKY2.T1.05, TKY2.T1.08, TKY2.T1.10, TKY2.T1.12, TKY2.T1.13, TKY2.T1.15, TKY2.T1.17, TKY.T1.01, TKY.T1.02, TKY.T2.02, TKY.T2.06, TKY.T2.07, TKY.T2.08 and TKY.T2.09 X-aptamers were significantly high binding affinity against target peptide. When we compared the binding affinity of GHRH NH2 (1–44) x-aptamers with GHRH NH2 (1–29) peptide, similar binding potential illustrated by GHRH NH2 (1–44) X-aptamers against GHRH (1–44) peptide in dot-blot analysis. Although aptamers against luteinizing hormone [34], thyroxine hormone [35], cortisol hormone [36] synthesized and characterized, there is not any study on GHRH aptamer synthesis. Thus, this is the first study reporting the X-aptamer synthesizing against both GHRH NH2 (1–44) and GHRH NH2 (1–29) peptide and also characterize each synthesized GHRH X-aptamers by dot-blot, immunofluorescence and, SPR analysis.

To evaluate the efficiency of aptamer and the incline impact of aptamers against their target molecules, the selection of appropriate targets is one of the most important steps in the aptamer selection process [29]. Generally, the dot-blot technique is used to determine the $K_d$ levels of aptamers with ligands [32]. Besides the dose-dependent aptamer-ligand dot blot analysis,

SPR analysis is also preferred to determine the dissociation fold of aptamers. From this point of view, for each putative X-aptamer dose-dependent dot-blot performed. Following Graph-Pad Prism non-linear regression analysis, TKY2.T1.01, TKY2.T1.02, TKY2.T1.03, TKY2.T1.04, TKY2.T1.05, TKY2.T1.08, TKY2.T1.11, TKY2.T1.13, TKY2.T1.15, TKY2.T1.17, TKY.T1.03, TKY.T1.04 X-aptamers has 21.60, 19.42, 21.45, 32.91, 139, 31.18, 28.91, 41.07, 131.3, 7.974, 46.40, 101.5, 21.60, 19.42, 21.45, 32.91, 5.139, 31.18, 28.91, 41.07, 131.3, 7.974, 46.40, 101.5 nM $K_d$ levels, respectively (Fig 3c). To confirm the results of the dot-blot technique, SPR analysis was performed for selected three putative aptamers (TKY2.T1.13, TKY.T1.08, and TKY.T1.09). SPR mediated $K_d$ determination was generally preferred for illustrating the apta-mer-ligand binding affinity [37]. In our study, three novel putative X-aptamers against GHRH were TKY2.T1.13, TKY.T2.08, TKY.T2.09 before SPR analysis (Fig 3f). Even if, aptamers are used as a therapeutic agent, the stability of aptamers under specific conditions such as temper-ature and nuclease-containing conditions. Generally, the serum stability of aptamers was eval-uated by mixing an equal volume of aptamer and human serum in a time-dependent manner [38]. TKY2.T1.02, TKY2.T1.03, TKY2.T1.04, TKY2.T1.05, TKY2.T1.13, TKY2.T1.15, TKY2.T1.17, TKY.T1.05, TKY.T2.02, TKY.T2.07, TKY.T2.08, TKY.T2.09 X-aptamers serum stability was prolonged to 120 h time points (Fig 3e). From so on, when we combine the $K_d$ level, co-immunofluorescence profile, and putative secondary structure, TKY2.T1.08, TKY2.T1.13, TKY.T1.08, and TKY.T1.09 X-aptamers selected for the competitive targeting of GHRH-GHRHR or GHRH-aptamer heterocomplexes. Co-immunofluorescence techniques are generally preferred for the binding profile of two ligands. As our putative X-aptamers were synthesized with 5'-biotinylated labeling, we can use streptavidin Alexa Fluor 488 to demon-strate the cell surface binding of X-aptamers on MIA PaCa-2 pancreatic cancer cell line (Fig 5). Co-immunofluorescence results illustrate the membrane binding of TKY2.T1.08, TKY2.T1.13, and TKY.T2.08 X-aptamers starting from the 50 nM concentrations. Concomitantly, this effect was increased by dose-dependent X-aptamers. However, clear fluorescence intensity was performed following 100 nm TKY.T2.09 treatment in MIA PaCa-2 cell by immunofluo-rescence staining. These results guide us to confirm the GHRH binding with GHRHR simulta-neously with cell surface attachment of X-aptamers. According to co-immunofluorescence results, merge figures illustrate the aptamer binding with GHRH on the surface of MIA PaCa-2 cells. Especially, a significant binding profile was detected for TKY2.T1.08, TKY.T2.08, and TKY.T2.09 only after 250 nM aptamer treatment. Following the binding affinity of GHRH X-aptamers against GHRH secreted from MIA PaCa-2 cells, we demonstrate the blocking impact of GHRH signaling *via* cAMP levels and GH/GHRHR expression profile. Due to cAMP assay kit results, both TKY2.T1.08 and TKY2.T1.13 X-aptamers declined cAMP levels compared to untreated controls in HT-29 colon cancer cells. In MIA PaCa-2 cells, both TKY.T2.08 and TKY.T2.09 X-aptamers decreased cAMP levels (Fig 6a). Moreover, each selected X-aptamer treatment sharply decreased the expression of both GH and GHRHR expressions as compared to untreated control cells in MIA Paca-2 cells (Fig 6b). Thus, TKY2.T1.08, TKY2.T1.13, TKY.T208, and TKY.T209 X-aptamers block GHRH signaling through suppression cAMP levels and inhibition on GH/GHRHR expression profiles. Finally, further studies may need to figure out the binding site of TKY2.T1.08, TKY2.T1.13, and TKY.T2.08 aptamers on GHRH epitope regions and determine the docking site for GHRHR and GHRH during GHRH-aptamer com-plex formation. Besides, *in silico* analysis, *in vitro* analysis might highlight the inhibitive effect of TKY2.T1.08, TKY2.T1.13, and TKY.T2.08, TKY.T2.09 X-aptamers on GHRH signaling.

In conclusion, this is the first report demonstrating the synthesis, selection, and characteri-zation of four putative X-aptamers against GHRH peptide. Although various peptide antago-nists targeting GHRH developed, there is not any X-aptamer targeting GHRH. We synthesized anti-GHRH aptamers by using X-aptamer technology *via* targeting both GHRH

NH2 (1–44) and NH2 (1–29). Due to new generation sequencing, 24 putative X-aptamer against GHRH synthesized. Among these x-aptamers, TKY2.T1.01, TKY2.T1.02, TKY2.T1.03, TKY2.T1.04, TKY2.T1.05, TKY2.T1.12, TKY2.T1.13, TKY.T1.01, TKY.T1.02, TKY.T2.07, TKY.T2.08, TKY.T2.09 were demonstrated to be a high binding affinity and low $K_d$ levels by dot-blot analysis. Due to serum stability and potential $K_d$ levels predictions; TKY2.T1.13, TKY.T2.08, TKY.T2.09 were selected for SPR analysis. In addition, novel two anti-GHRH NH2 (1–44) and two anti-GHRH NH2 (1–29) X-aptamers were characterized according to their binding ability on *in vitro* systems by using high GHRH expressing and secreting cell lines. Two anti-GHRH NH2 (1–29) x-aptamers have a high binding affinity, low $K_d$ levels, and elevated binding potential in low doses compared to anti-GHRH NH2 (1–44) x-aptamers. Similar to GHRH peptide antagonists, 1–29 amino acid residue of GHRH peptide is more essential for the targeting and synthesizing efficient GHRH binding aptamers. However, new studies might be performed to determine the potential inhibitive effect of these x-aptamers on GHRH signaling and the determination of docking sites for x-aptamers on GHRH ligands. In addition, further *in vitro*, *in vivo*, and *in silico* analysis may need to highlight the impact of these three-novel X-aptamers efficiencies on GHRH signaling.

## Supporting information

**S1 Raw images.**
(PDF)

## Acknowledgments

**Ethical statement:** This article does not involve any study with human participants or animals performed by any of the authors. The research performed on commercially available cell lines.

## Author Contributions

**Funding acquisition:** Ajda Coker-Gurkan.

**Investigation:** Ajda Coker-Gurkan.

**Methodology:** Burcu Ayhan-Sahin, Zeynep-Elif Apaydın, Pınar Obakan-Yerlikaya, Elif-Damla Arisan, Ajda Coker-Gurkan.

**Supervision:** Ajda Coker-Gurkan.

**Validation:** Burcu Ayhan-Sahin, Pınar Obakan-Yerlikaya, Elif-Damla Arisan.

**Writing – original draft:** Pınar Obakan-Yerlikaya, Elif-Damla Arisan, Ajda Coker-Gurkan.

**Writing – review & editing:** Pınar Obakan-Yerlikaya, Elif-Damla Arisan, Ajda Coker-Gurkan.

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
