## [Decision Letter · Decision Letter 0]

3 Mar 2021

PONE-D-21-00439

Synthesis and characterization of novel ssDNA X-Aptamers Targeting Growth Hormone Releasing Hormone (GHRH)

PLOS ONE

Dear Dr. GURKAN,

Thank you for submitting your manuscript to PLOS ONE. After careful consideration, we feel that it has merit but does not fully meet PLOS ONE’s publication criteria as it currently stands. Therefore, we invite you to submit a revised version of the manuscript that addresses the points raised during the review process.

1) It is missing some importante controls (blockage of GHRH signalling, competitor in excess concentration  to correct for unspecific binding at all aptamer doses, negative controls of the imune reactions);

2) Please, see the comments raised by both the reviewers.

We look forward to receiving your revised manuscript.

Kind regards,

Paulo Lee Ho, Ph.D.

Academic Editor

PLOS ONE

Journal Requirements:

2)  Thank you for stating the following in the Acknowledgments Section of your manuscript:

[The project was funded by the Scientific and 425 Technological Research Council of Turkey (TUBITAK) (Project Number: 117Z254).]

 [The funders had no role in study design, data collection and analysis, decision to

publish, or preparation of the manuscript.]

3) We note that you have stated that you will provide repository information for your data at acceptance. Should your manuscript be accepted for publication, we will hold it until you provide the relevant accession numbers or DOIs necessary to access your data. If you wish to make changes to your Data Availability statement, please describe these changes in your cover letter and we will update your Data Availability statement to reflect the information you provide.

4) PLOS ONE now requires that authors provide the original uncropped and unadjusted images underlying all blot or gel results reported in a submission’s figures or Supporting Information files. This policy and the journal’s other requirements for blot/gel reporting and figure preparation are described in detail at https://journals.plos.org/plosone/s/figures#loc-blot-and-gel-reporting-requirements and https://journals.plos.org/plosone/s/figures#loc-preparing-figures-from-image-files. When you submit your revised manuscript, please ensure that your figures adhere fully to these guidelines and provide the original underlying images for all blot or gel data reported in your submission. See the following link for instructions on providing the original image data: https://journals.plos.org/plosone/s/figures#loc-original-images-for-blots-and-gels.

5) Please amend your list of authors on the manuscript to ensure that each author is linked to an affiliation. Authors’ affiliations should reflect the institution where the work was done (if authors moved subsequently, you can also list the new affiliation stating “current affiliation:….” as necessary).

Reviewers' comments:

Reviewer's Responses to Questions

**Comments to the Author**

1. Is the manuscript technically sound, and do the data support the conclusions?

Reviewer #1: Yes

Reviewer #2: Partly

2. Has the statistical analysis been performed appropriately and rigorously? 

Reviewer #1: Yes

Reviewer #2: Yes

3. Have the authors made all data underlying the findings in their manuscript fully available?

Reviewer #1: Yes

Reviewer #2: No

4. Is the manuscript presented in an intelligible fashion and written in standard English?

Reviewer #1: Yes

Reviewer #2: Yes

5. Review Comments to the Author

Reviewer #1: This work focuses on the synthesis and purification of X-aptamers against Growth Hormone Releasing Hormone (GHRH), which was expressed in HEK293 and bacteria cells. GHRH is a hypothalamic neuropeptide that stimulates the production of growth factor GH, which in turn stimulates cell proliferation and division in different types of cancer.

24 X-aptamers were synthesized and 3 of them (TKY1.T1.13, TKY.T2.08, TKY.T2.09) had the best stability in serum and the highest binding potential by GHRH with low levels of Kd.

The work carried out is original, it has great relevance. It is well written and in my opinion meets the requirements to be accepted. However, some minor details need to be corrected.

>the quality of the figures must be corrected

>figure 2E does not match between title and file. There is no mention of this figure in the text. It also seems unnecessary to me since it is redundant with the 3D figure

>Figure 5a-5h should be considered as supplementary material since the main purpose of the work is not to determine the ability of aptamers to bind to the cell lines studied

>Additional experiments are necessary to determine the ability of the obtained aptamers to block GHRH signaling. or at least make it clear that these experiments were not done (Page 19, lines 304-305 it is mentioned that aptamers are used to block GHRH).

>make clear the limitations of the study

Reviewer #2: The authors report the development of aptamers selectively binding to the GHRH peptide and show binding efficiency by dot blot and SPR assays as well as by immunofluorescence assays with cells.

Issues to be addressed:

1. Overall: Image quality is low

2. Where should be an explanation for X-aptamers, and not just a reference. Can X aptamer sequences by modeled by using the mFOLD software?

3. How many SELEX cycles were performed? Why so many PCR cycles were performed, which indeed lead to the accumulation of PCR artefacts and not of selected aptamers. Was there any problem with the efficiency of the PCR reaction? The sequenced aptamers have a long 5` terminal T extension, I would guess that this an artefact. I suggest to align the aptamer consensus sequences in Figure 2C and make sure, which sequences belong to constant and variable sequences of the aptamers. Figure 2D should show, where in the loop region the consensus sequences can be found, and most importantly, the sequences should be readable. Figure 2E shows the stability of DNA aptamers along time. As this figure stands, it is not worth anything. First DNA needs to be resolved in a high-resolution denaturing polyacrylamide gel (which could distinguish between different sizes of aptamers); second, I do not see any positive controls for this assay (i.e. nuclease, which should degrade the aptamers).

4. The following experiment has a problem: "Then binding affinity of X-aptamers to target was assessed in increasing aptamer doses. Fold of change was calculated according to the dot intensities and dissociation constant (Kd) was calculated by nonlinear regression". Such binding data contain specific and unspecific binding. A competitor in excess concentration needs to be used in order to correct for unspecific binding at all aptamer doses.

5. A two or three of change of binding is very little, assuming that the unselected aptamer library shows unspecific binding in the mM range. Binding affinities of aptamers should increase at least 1,000 fold, reflected by more than of a two or three fold amount of target bound aptamers.

6. Negative controls of immune reactions, including those with aptamers are missing.

6. PLOS authors have the option to publish the peer review history of their article (what does this mean?). If published, this will include your full peer review and any attached files.

Reviewer #1: **Yes: **Ospina-Villa Juan David

Reviewer #2: No

---

## [Author Response · Author response to Decision Letter 0]

15 Apr 2021

Review Comments to the Author

Reviewer #1: This work focuses on the synthesis and purification of X-aptamers against Growth Hormone Releasing Hormone (GHRH), which was expressed in HEK293 and bacteria cells. GHRH is a hypothalamic neuropeptide that stimulates the production of growth factor GH, which in turn stimulates cell proliferation and division in different types of cancer.

24 X-aptamers were synthesized and 3 of them (TKY1.T1.13, TKY.T2.08, TKY.T2.09) had the best stability in serum and the highest binding potential by GHRH with low levels of Kd.

The work carried out is original, it has great relevance. It is well written and in my opinion meets the requirements to be accepted. However, some minor details need to be corrected.

Comment 1: the quality of the figures must be corrected.

Response to comment : All the figures quality is corrected

Comment 2 : figure 2E does not match between title and file. There is no mention of this figure in the text. It also seems unnecessary to me since it is redundant with the 3D figüre

Response to comment : According to reviewer’s comment about the figure 2e that was not matched with the title the mentioned figure was discarded from the revised version of the manuscript. 

Comment 3: Figure 5a-5h should be considered as supplementary material since the main purpose of the work is not to determine the ability of aptamers to bind to the cell lines studied

Response to comment : According to other reviewer’s comment about figure 5a-5h, these figures couldn’t substituted to supplementary material. We performed GHRH siRNA transfection assays for this figure in order to demonstrate the effective binding of aptamers with GHRH ligand. 

Comment 4: Additional experiments are necessary to determine the ability of the obtained aptamers to block GHRH signaling. or at least make it clear that these experiments were not done (Page 19, lines 304-305 it is mentioned that aptamers are used to block GHRH).

Response to comment : According to reviewer’s comment, the cAMP assay and coimmunofluorescence assays were performed. As the major downstream target of GHRH signaling were cAMP levels and target gene expressions (GH, GHRHR), we selected these experiments. Due to cAMP assay, significant decrease in cAMP levels were observed following selected aptamer treatment in HT29 and MIA Paca2 cells. Moreover, both GH and GHRHR expression profile after aptamers treatment was determined in MIA Paca-2 cells. All the results were inserted in the revised manuscript. 

Comment 5: make clear the limitations of the study

Response to comment : In our previous version we tried to make clear the limitations of study by giving as “Finally, further studies may need to figure out the binding site of TKY2.T1.08, TKY2.T1.13, and TKY.T2.08 aptamers on GHRH epitope regions and determine the docking site for GHRHR and GHRH during GHRH-aptamer complex formation. Besides, in silico analysis, in vitro analysis might highlight the inhibitive effect of TKY2.T1.08, TKY2.T1.13, and TKY.T2.08, TKY.T2.09 X-aptamers on GHRH signaling.” And also in conclusion part we added “However, new studies might be performed in order to figure out the potential inhibitive effect of these x-aptamers on GHRH signaling and determination of docking sites for x-aptamers on GHRH ligands. In addition, further in vitro, in vivo and in silico analysis may need to highlight the impact of these three-novel x-aptamers efficiency on GHRH signaling.” 

Reviewer #2: The authors report the development of aptamers selectively binding to the GHRH peptide and show binding efficiency by dot blot and SPR assays as well as by immunofluorescence assays with cells.

Issues to be addressed:

Comment 1. Overall: Image quality is low

Response to comment : Low quality images were corrected. 

Comment 2. Where should be an explanation for X-aptamers, and not just a reference. Can X aptamer sequences by modeled by using the mFOLD software?

Response to comment : According to reviewer’s comment, instead of giving explanation for X-aptamer, we gave also reference in the revised version of the manuscript. Due to first reviewer the secondary structure of aptamers is not linked with the title so we discarded the 3D structure figures. 

Comment 3. How many SELEX cycles were performed? Why so many PCR cycles were performed, which indeed lead to the accumulation of PCR artefacts and not of selected aptamers. Was there any problem with the efficiency of the PCR reaction? The sequenced aptamers have a long 5` terminal T extension, I would guess that this an artefact. I suggest to align the aptamer consensus sequences in Figure 2C and make sure, which sequences belong to constant and variable sequences of the aptamers. Figure 2D should show, where in the loop region the consensus sequences can be found, and most importantly, the sequences should be readable. Figure 2E shows the stability of DNA aptamers along time. As this figure stands, it is not worth anything. First DNA needs to be resolved in a high-resolution denaturing polyacrylamide gel (which could distinguish between different sizes of aptamers); second, I do not see any positive controls for this assay (i.e. nuclease, which should degrade the aptamers).

Response to comment : Due to X-aptamer synthesis kit instructions we performed increasing number of PCR amplifications in order to increase the yield of aptamers that are specific to GHRH peptide. So we performed 14, 22, 25 cycle of binding the magnetic bead based library with GHRH peptide. As this is a patented technology for generation of X-aptamers by kit based selection so accumulation of PCR artifacts were get rid of by each step washing procedure as every PCR cycle product was used as a template library for each following SELEX step. (https://www.youtube.com/watch?v=CgfX-KmXdA0). After amplifications, AM Biotechnologies were generated NGS in order to determine the novel putative sequences instead the ones similar with the 5’ and 3’ library linker sequences. (https://www.genomeweb.com/resources/new-product/am-biotechnologies-x-aptamer-selection-kit). All the sequences aligned and T rich 5’ region with “CACGAC” repeats were named as 5’ constant sequence and “GTGGGGCCCATG” sequence repeat at the 3’ region were named as 3’ constant sequence. The sequences between these fixed sequences were named as “variable sequence” . For the figure 2d, according to valuable contribution, we changed the figure to more readable one in the revised version of the text. According to comment on figure 2e, we performed polyacrylamide gel electrophoresis for serum stability of each aptamer and changed all the figures with the new results. Moreover, in each serum stability gel results, we added positive control as DNase I digested aptamers. 

Comment 4. The following experiment has a problem: "Then binding affinity of X-aptamers to target was assessed in increasing aptamer doses. Fold of change was calculated according to the dot intensities and dissociation constant (Kd) was calculated by nonlinear regression". Such binding data contain specific and unspecific binding. A competitor in excess concentration needs to be used in order to correct for unspecific binding at all aptamer doses.

Response to comment : According to valuable contribution, we performed GHRH sandwich ELISA assay and dot blot analysis in order to detect the specifity of x-aptamers in dose-dependent manner for selected aptamers. The performed experiment results were given in Figure 3d. As one of most effective competitor for aptamer is antibody, we preffer to perform GHRH ELISA assay. GHRH ELISA is a sandwich ELISA and wells were coated with GHRH antibody. 500 nM GHRH peptide is applied to wells and dose-dependent aptamers was used as they are 5’ biotinylated in order to determine the colorimetric analysis. And we performed dot-blot from supernatant of GHRH-aptamer complex in dose-dependent manner for selected x-aptamers. Thus, we demonstrated dose-dependent x-aptamer is a competitor for GHRH antibody and x-aptamers bind to target GHRH peptide with a high binding affinity. 

Comment 5. A two or three of change of binding is very little, assuming that the unselected aptamer library shows unspecific binding in the mM range. Binding affinities of aptamers should increase at least 1,000 fold, reflected by more than of a two or three fold amount of target bound aptamers.

Response to comment : According to valuable contribution, we repeated dot-blot analysis for all aptamers. And then calculated the fold of change due to colorimetric analysis of each spot comparing with the negative control as only GHRH ligand was used. The repeated experiment results and fold of change in GHRH binding affinity due to dot-blot analysis was replaced in the revised version of the manuscript. 

Comment 6. Negative controls of immune reactions, including those with aptamers are missing.

Response to comment : According to valuable contribuition, we applied GHRH siRNA as a negative control of immune reactions. Beside dose-dependent aptamer applications, we also apply aptamers in MIA Paca-2 cells with blocked GHRH signaling by siRNA. Due to experiments GHRH silencing as a negative control for immune reactions demonstrated that aptamers didn’t bind to MIA Paca-2 cells. The new experiment results were replaced in the figure 5b-d-f-h in the revised version of the figures.

---

## [Decision Letter · Decision Letter 1]

10 Jun 2021

PONE-D-21-00439R1

Synthesis and characterization of novel ssDNA X-Aptamers Targeting Growth Hormone Releasing Hormone (GHRH)

PLOS ONE

Dear Dr. GURKAN,

Thank you for submitting your manuscript to PLOS ONE. After careful consideration, we feel that it has merit but does not fully meet PLOS ONE’s publication criteria as it currently stands. Therefore, we invite you to submit a revised version of the manuscript that addresses the points raised during the review process.

We look forward to receiving your revised manuscript.

Kind regards,

Paulo Lee Ho, Ph.D.

Academic Editor

PLOS ONE

Reviewers' comments:

Reviewer's Responses to Questions

**Comments to the Author**

1. If the authors have adequately addressed your comments raised in a previous round of review and you feel that this manuscript is now acceptable for publication, you may indicate that here to bypass the “Comments to the Author” section, enter your conflict of interest statement in the “Confidential to Editor” section, and submit your "Accept" recommendation.

Reviewer #1: All comments have been addressed

Reviewer #2: (No Response)

Reviewer #3: (No Response)

2. Is the manuscript technically sound, and do the data support the conclusions?

Reviewer #1: Yes

Reviewer #2: No

Reviewer #3: Yes

3. Has the statistical analysis been performed appropriately and rigorously? 

Reviewer #1: Yes

Reviewer #2: No

Reviewer #3: Yes

4. Have the authors made all data underlying the findings in their manuscript fully available?

Reviewer #1: Yes

Reviewer #2: Yes

Reviewer #3: Yes

5. Is the manuscript presented in an intelligible fashion and written in standard English?

Reviewer #1: Yes

Reviewer #2: No

Reviewer #3: Yes

6. Review Comments to the Author

Reviewer #1: All comments were resolved and incorporated in the new version of the manuscript. Additional experiments were carried out that clearly dispel the initial doubts of the reviewers

Reviewer #2: Authors showed some new data, but the paper remains unacceptable for publication. Besides the authors did not respond satisfactorily to my queries, the manuscript is full of grammar errors, making it almost impossible to understand the messages, which the authors would like to transmit.

Examples for poor English, sometimes together with conceptional errors.

“x-aptamers was used as a detection antibody in order to detect GHRH peptide that applied on GHRH antibody coated plates. “ An aptamer is not an antibody

“To confirm the ELISA results, we performed dot-blot analysis of upper phase of GHRH peptide 296 and X-aptamer combination in dose-dependent manner by dot-blot analysis. “

“As there is not any competitor for GHRH X-aptamers except antibody against GHRH peptide, we performed GHRH ELISA 298 assay. “ Not true, unlabeled aptamers in excess can be used to compete with binding of labeled aptamers.

“According to the serum stability results, TKY2.T1.02, TKY2.T1.04, TKY2.T1.05, 304 TKY2.T1.08, TKY2.T1.13, TKY2.T1.17, TKY.T2.05, TKY.T1.05, TKY.T2.02, TKY.T2.07, TKY.T2.08 and TKY.T2.09 X-aptamers were seems to be stable in human serum within 120 h period at 37o “

“X-aptamers were significantly deplete intracellular cAMP 338 concentration as compared to untreated control cells in HT-29 cells. When we checked the results, no significant effect was observed for cAMP levels for TKY2.T1.08 and TKY2.T1.13 X-aptamers application 340 in HT-29 cells“. Contradictory!

Scientific issues:

PCR cycles: I did not mean SELEX cycles, but PCR cycles needed for amplification of recovered DNA. If i.e. 1 % of the material was recovered following selection, a number of PCR cycles needed to restore 100% can be calculated and certainly will not give 25 cycles.

I know that the technique is patented, but nether the less technical details need to be given, that any researcher can repeat experiments.

Regression plots in Figure 3 are not readable and they seem to rely on a single measurement.

I asked the authors to provide an alignment of variable sequences, which allows the definition of structural classes. Authors just listed the sequences without any bioinformatical analysis.

I asked the authors to adequately introduce X-DNA aptamers (why authors select X-aptamers). This should be done in more detail and not by just citing a reference. The message is confusingly diluted along the paper. However, it should be clearly cited in the introduction.

Reviewer #3: The anuscript revision was attached as a word doc file, since it exceeded the limit of 20000 characters

7. PLOS authors have the option to publish the peer review history of their article (what does this mean?). If published, this will include your full peer review and any attached files.

Reviewer #1: **Yes: **Ospina-Villa Juan

Reviewer #2: No

Reviewer #3: No

---

## [Author Response · Author response to Decision Letter 1]

8 Jul 2021

Introduction

The study presents the synthesis of 24 putative X-aptamers against GHRH peptides with high serum stability, target binding, and low Kd levels.

Overall: 

English needs proofreading. 

- Manuscript English was corrected and some sentences were rewritten. 

Figure legends should be more informative, otherwise, the reader has to look back and forth to find the information scattered in the manuscript. 

- Figure legends were detailed and altered to more informative. 

Specific questions: 

Comment#1 Introduction does not present clear information regarding the goal of making those aptamers. Do authors want to make GHRH antagonists? Authors want to make new therapeutic agents or diagnostic tools? It’s not clear and should be clarified in the Introduction section. 

Reply to comment 1: According to valuable comment, we explained in the introduction part that we want to make GHRH aptamer antagonists. And we added “In order to generate GHRH antagonists, we select aptamers due to their high binding affinity, low toxicity, non-immunogenic properties (11)” sentense after GHRH peptide antagonists in the revised text. 

Comment#2 Why authors want to make aptamers against 1-44 and/or 1-29 peptides? Are there any advantages/differences of using each peptide target for selection? A better explanation in the Introduction regarding the reasons for testing both targets is desirable. The only thing that is mentioned is the fact that biological activity is observed in the 1-29 sequence. If so, why not only 1-29 protein was used as a target? Also, I could not find in the Discussion section, whether 1-44 targeting aptamers were also able to detect 1-29 protein and vice versa. I believe this should be addressed in the discussion. 

Rept to comment 2: We rewrote the introduction and discussion parts to explain why we use both peptides for aptamer synthesis. This is the first report demonstrating the synthesis of aptamers against GHRH so we want to generate SELEX against both 1-44 and 1-29. Like the same strategy with Macugen (anti-VEGF), we preferred to synthesize X-aptamers against GHRH full length protein which is 44 amino acid long. Moreover, this will give us to detect every epitopic part of GHRH variants when we synthesized aptamers. In addition, due to GHRH peptide antagonist that targets GHRH 1-29, we used 1-29 peptide as a target to demonstrate the impact of aptamers by comparing every epitopic region. We inserted “As there is not any GHRH antagonist aptamer, we preferred to synthesize aptamers against both GHRH 1-44 and 1-29. In order to synthesize X-aptamers that can capture every epitopic region, we selected full GHRH protein NH2 (1-44). Besides, as GHRH peptide antagonist select 1-29 region, we used GHRH NH2 (1-29) as a target in order to synthesize X-aptamers. In order to illustrate by comparing the binding affinity of GHRH antagonist X-aptamers, we aimed to synthesize, select X-aptamers against both GHRH NH2 (1-44) and NH2 (1-29) peptides.” Due to dot blot analysis, both GHRH NH2 (1-44) X-aptamers and GHRH NH2 (1-29) aptamers have similar binding affinity against GHRH peptide target. Thus, in discussion section we added “When we compared the binding affinity of GHRH NH2 (1-44) x-aptamers with GHRH NH2 (1-29) peptide, similar binding potential illustrated by GHRH NH2 (1-44) X-aptamers against GHRH (1-44) peptide in dot-blot analysis.”

Comment#3 X-aptamer technology was recently developed. The advantages and limitations of this SELEX procedure need to be further explored in the Introduction section. 

Reply to comment 3: The advantages and limitations of this SELEX procedure were given in the introduction part as “The most important advantage of X-aptamer technology is to synthesize up to 5 different target at the same time by one SELEX method. However, the limitation of X-aptamer technology is molecular size of target molecule, targets longer than 10 amino acids is assumed to be more preferred (14).”

Comment#4 Line 99, “Expression of bacterial His” needs to be further explained. Is it the control of the experiment? Also, plasmids pCMV3-SP-N-His-NCV and pCMV3-SP-His-ORF His-GHRH are not defined. Which proteins are coded by these plasmids? A detailed explanation of each construct is desirable. 

Rept to comment 4: The expression of bacterial His performed as a control in order to determine the His-tagged protein expression in HEK293 cells. His tagged protein synthesized by pCMV3-SP-N-His-NCV plasmid and His-tagged GHRH peptide expressed from pCMV3-SP-His-ORF His-GHRH plasmid. This explanation was given in the revised version of the manuscript. 

Comment#5 Line 160. What is the composition of the selection buffer?

Reply to comment 5: The composition of the selection buffer is given in the revised version of the manuscript. The content of selection buffer is 1X PBS pH:7.4, 1 mM MgCl2, 0.05% Tween20, 0.02% BSA.

Comment#6 Line 173 “4 1.5 ml tubes”…needs improvement

Reply to comment 6: “4 1.5 ml tubes” was changed to “four 1.5 mL tubes” in the revised manuscript.

Comment#7 Units are wrong throughout the whole manuscript ….for example μl needs to be replaced by μL, ml by mL, etc.

Reply to comment 7: We changed them all in the revised text. 

Comment# 8 Line 248-249. The sentence appears incomplete.

Reply to comment 8: According to valuable comment, the sentence appears to be incomplete was changed to “Purification of His-tagged GHRH peptide “ as a title for result section. 

Comment#9 Line 253…” HEK293 cells were used…” 

Reply to comment 9: According to valuable comment, “HEK293 cells were used” changed the revised manuscript. 

Comment#10 Methodology of aptamer selection is very confusing. Author state: “Our aim in this study is to synthesize and purify GHRH (1-44) protein in HEK293 cells, synthesize and select X-aptamers against GHRH peptide both 1-29 and 1-44, and demonstrate synthesized aptamers’ target binding activity as well as serum stability.” 

If only purification of GHRH 1-44 protein was performed, how the selection was performed using both proteins (1-29 and 1-44) as targets? There were two different Selex procedures, each one using one protein as the target? That is not clear. If authors claim that will select aptamers against both 1-29 and 1-44 peptides, the methodology should provide a detailed description of which peptide was used target for selection. 

Also, seems strange that authors tried to express in E.coli since it’s widely known that bacteria lack a posttranscriptional modification mechanism. Line 252-254 says “Since the bacterial protein synthesis mechanism lacks post-translational modifications, HEK293 cells were used…” this information was already available before the bacterial expression was performed in this article, therefore, I wonder why bacterial expression was tested and included in the study… 

Although using HEK293 for protein expression, it is not clear if 1-29 or 1-44 peptides were expressed. 

Reply to comment 10: We realized that we didn’t write in the material method section that we used bacterial and eukaryotic GHRH NH2 (1-44) peptide and also GHRH NH2 (1-29) peptide as a target for X-aptamer SELEX. We performed two SELEX; first SELEX; bacterial/eukaryotic SELEX (Figure 2a) and second SELEX: GHRH 1-29 (figure 2b). As X-aptamer synthesing kit targets up to 5 targets, we used purified bacterial and eukaryotic GHRH 1-44 peptide for the first SELEX and also GHRH 1-29 peptide for the second SELEX. According to manufacturers instructions, target must be magnetically labelled with either his or streptavidin magnetic beads. Thus, we have to use His-tagged GHRH protein for his-magnetic beads and also biotinylated GHRH for streptavidin magnetic bead. Thus, in order to compare the binding affinity of X-aptamers against each target we prefer to use GHRH 1-44 and GHRH 1-29 peptides whether first 1-29 peptide or full length peptide epitopic regions determine the binding affinity of X-aptamers. To synthesize aptamers against GHRH 1-44 full length target, we used both E.coli HB101 cells to gain high yield of His-GHRH protein. However to compare the binding affinity of x-aptamers against bacterial and eukaryotic GHRH 1-44 peptide, we performed purification from each host. In addition, as HEK293 cells has no expression for GHRH, they express what we transfected through a plasmid and the plasmid has full length GHRH peptide with His-tagged. Thus, HEK293 cells express GHRH 1-44 peptide and this was confirmed by the immunoblotting results. 

Comment#11 Sequences provided in Figure 2 are aptamers obtained from which selection (1-29 or 1-44)? What is the difference between sequences named TKY2 and TKY? 

Reply to comment 11: The sequences provided in Figure 2 are aptamers obtained from both bacterial/eukaryotic GHRH 1-44 and GHRH 1-29 peptides. Names for TKY2 belongs to bacterial/eukaryotic GHRH 1-44 peptide and TKY belongs to GHRH 1-29 peptide. The detailed explanation was given as “Aptamers against bacterial His-tagged GHRH were TKY2.T1.01, TKY2.T1.02, TKY2.T1.03, TKY2.T1.04, TKY2.T1.05, TKY2.T1.08, TKY2.T1.10, TKY2.T1.12, TKY2.T1.13, TKY2.T1.15, TKY2.T1.17. TKY.T1.01, TKY.T1.02, TKY.T1.03, TKY.T1.04, TKY.T1.05 aptamers were against eukaryotic GHRH NH2 (1-44) peptide. Putative X-aptamers against GHRH 1-29 target named as TKY.T2.02, TKY.T2.06, TKY.T2.07, TKY.T2.08, TKY.T2.09.” in the results section of the revised manuscript. 

Comment#12 Figure 1 shows expression and purification of NH2 (1-44) target or GHRH NH2 (1-29) target? Figure legends should be more informative, otherwise, the reader has to look back and forth to find the information scattered in the manuscript. 

Reply to comment 12: We only purified GHRH 1-44 from bacterial and eukaryotic host origin, we bought the GHRH 1-29 peptide from Sigma G6771 as a product. Thus, the figure 1 represents the expression and purification of GHRH 1-44 target. It is mentioned in the figure legend 1 in the revised manuscript. 

Comment#13. Bacterially expressed His-GHRH protein was expressed as inclusion bodies and denatured using Guanidine. No protocol for protein refolding was presented.

Reply to comment 13: Protein refolding by bacterial expressed His-GHRH protein as inclusion bodies and denatured using Guanidine, we gave reference in the revised manuscript. (https://www.ncbi.nlm.nih.gov/pmc/articles/PMC2324003/)

Review Comments to the Author

Reviewer #1: All comments were resolved and incorporated in the new version of the manuscript. Additional experiments were carried out that clearly dispel the initial doubts of the reviewers

Reviewer #2: Authors showed some new data, but the paper remains unacceptable for publication. Besides the authors did not respond satisfactorily to my queries, the manuscript is full of grammar errors, making it almost impossible to understand the messages, which the authors would like to transmit.

Examples for poor English, sometimes together with conceptional errors.

- All the grammar errors were corrected, manuscript English is proofread and some sentences were rewritten again in order to transmit the messages. 

Comment 1: “x-aptamers was used as a detection antibody in order to detect GHRH peptide that applied on GHRH antibody coated plates. “ An aptamer is not an antibody

Reply to comment: We changed the sentence as “ According to GHRH ELISA results, instead of using detection antibody, we used x-aptamers to competitively capture GHRH peptide. “ We know that an aptamer is not an antibody but as we dont have any unlabelled aptamers we prefered to use GHRH competitive ELISA in order to demonstrate the specificity of x-aptamers through their binding affinity against target molecule. 

Comment 2: “To confirm the ELISA results, we performed dot-blot analysis of upper phase of GHRH peptide 296 and X-aptamer combination in dose-dependent manner by dot-blot analysis.

Reply to comment 2: According to valueable contribution, we changed the sentence as “To confirm the ELISA results, we performed dot-blot analysis in dose-dependent manner“.

Comment 3: “As there is not any competitor for GHRH X-aptamers except antibody against GHRH peptide, we performed GHRH ELISA 298 assay. “ Not true, unlabeled aptamers in excess can be used to compete with binding of labeled aptamers.

Reply to comment 3: According to valueable contributions, we changed the sentence as “As we do not have any unlabeled aptamers, GHRH antibody is used as a competitor for GHRH X-aptamers against GHRH peptide binding“. We know unlabelled excess aptamer will be used for competitive for GHRH binding to determine specifity of aptamers. However, we synthesized only 5’ biotin labeled aptamers from AM Biochemicals. As we couldn’t purchase both labelled and unlabelled 24 putative aptamers, we can only demonstrate by competitive ELISA Assay in dose-dependent manner. 

Comment 4: “According to the serum stability results, TKY2.T1.02, TKY2.T1.04, TKY2.T1.05, 304 TKY2.T1.08, TKY2.T1.13, TKY2.T1.17, TKY.T2.05, TKY.T1.05, TKY.T2.02, TKY.T2.07, TKY.T2.08 and TKY.T2.09 X-aptamers were seems to be stable in human serum within 120 h period at 37o “

Reply to comment 4: We changed the sentence as “According to the serum stability results, TKY2.T1.02, TKY2.T1.04, TKY2.T1.05, TKY2.T1.08, TKY2.T1.13, TKY2.T1.17, TKY.T2.05, TKY.T1.05, TKY.T2.02, TKY.T2.07, TKY.T2.08 and TKY.T2.09 X-aptamers were stable in human serum within 120 h period at 37oC (Fig 3e).”

Comment 5: “X-aptamers were significantly deplete intracellular cAMP 338 concentration as compared to untreated control cells in HT-29 cells. When we checked the results, no significant effect was observed for cAMP levels for TKY2.T1.08 and TKY2.T1.13 X-aptamers application 340 in HT-29 cells“. Contradictory!

Reply to comment 5: The sentence was changed to “Due to cAMP assay analysis, both TKY2.T1.08 and TKY2.T1.13 X-aptamers significantly depleted intracellular cAMP concentration as compared to untreated control cells in HT-29 cells. When we checked the results, no significant effect observed for cAMP levels for TKY2.T1.08 and TKY2.T1.13 X-aptamers application in MIA Paca-2 cells. In addition, we measured statistically significant decline in cAMP levels following TKY.T2.08 and TKY.T2.09 X-aptamers treatment in MIA Paca-2 cells (Fig 6a).”

Scientific issues:

Comment 6: PCR cycles: I did not mean SELEX cycles, but PCR cycles needed for amplification of recovered DNA. If i.e. 1 % of the material was recovered following selection, a number of PCR cycles needed to restore 100% can be calculated and certainly will not give 25 cycles.

I know that the technique is patented, but nether the less technical details need to be given, that any researcher can repeat experiments.

Reply to comment 6: In the method, we do not aim to recover 100% DNA. Instead, the company requested the gel image and informed that it is sufficient for Next Generation Sequencing.

Comment 7: Regression plots in Figure 3 are not readable and they seem to rely on a single measurement.

Reply to comment 7: We changed figure as dot-blot analysis and also regression plots in order to increase its readability. The experiment was repeated three times and given figures illustrates the representative of one of the experiment. 

Comment 8: I asked the authors to provide an alignment of variable sequences, which allows the definition of structural classes. Authors just listed the sequences without any bioinformatical analysis.

Reply to comment 8: According to comment, we align the variable sequences by bioinformatic analysis and given in the revised version of the manuscript. 

Comment 9: I asked the authors to adequately introduce X-DNA aptamers (why authors select X-aptamers). This should be done in more detail and not by just citing a reference. The message is confusingly diluted along the paper. However, it should be clearly cited in the introduction.

Reply to comment 9: We explained in detail the X-aptamer technology, advantages and limitations of aptamers. Moreover, we clearly cited why we select X-aptamers in the introduction part. 

Reviewer #3: The manuscript revision was attached as a word doc file, since it exceeded the limit of 20000 characters

---

## [Decision Letter · Decision Letter 2]

26 Aug 2021

PONE-D-21-00439R2

Synthesis and characterization of novel ssDNA X-Aptamers Targeting Growth Hormone Releasing Hormone (GHRH)

PLOS ONE

Dear Dr. GURKAN,

Thank you for submitting your manuscript to PLOS ONE. After careful consideration, we feel that it has merit but does not fully meet PLOS ONE’s publication criteria as it currently stands. Therefore, we invite you to submit a revised version of the manuscript that addresses the points raised during the review process, specially those raised by reviewer #2.

We look forward to receiving your revised manuscript.

Kind regards,

Paulo Lee Ho, Ph.D.

Academic Editor

PLOS ONE

Reviewers' comments:

Reviewer's Responses to Questions

**Comments to the Author**

1. If the authors have adequately addressed your comments raised in a previous round of review and you feel that this manuscript is now acceptable for publication, you may indicate that here to bypass the “Comments to the Author” section, enter your conflict of interest statement in the “Confidential to Editor” section, and submit your "Accept" recommendation.

Reviewer #2: (No Response)

Reviewer #3: All comments have been addressed

2. Is the manuscript technically sound, and do the data support the conclusions?

Reviewer #2: (No Response)

Reviewer #3: Yes

3. Has the statistical analysis been performed appropriately and rigorously? 

Reviewer #2: No

Reviewer #3: N/A

4. Have the authors made all data underlying the findings in their manuscript fully available?

Reviewer #2: Yes

Reviewer #3: Yes

5. Is the manuscript presented in an intelligible fashion and written in standard English?

Reviewer #2: Yes

Reviewer #3: Yes

6. Review Comments to the Author

Reviewer #2: I agree that the authors made some progress, but I have yet serious reservations regarding the scientific quality of this manuscript.

1. Authors did not understand the point of my concerns of a large number of PCR cycles for reestablishing the X-DNA pool. The point is that over amplification, i.e. by 25 cycles, for reestablishing 100 % from 1 % results in a loss of pool and sequence heterogeneity and selection of artefacts. This is shown in the alignment analysis that authors now provided. It is expected that there are conserved consensus motifs within a part of the previous random region, but not that practically the entire random region makes part of the random regions. If this would be like that, how authors explain the differences in binding affinities.

2. Figure 3c: Where is the control (scrambled sequence aptamer) for these assays. Increase of an unspecific aptamer concentration will augment binding, however cannot be saturated. This needs to be shown. I understand that a non-linear curve using a zero point needs to be performed. Please, check for that TKY2.T1.T2, TKY.T1.01 and TKY.T1.02. The fittings are just wrong. Are these just one-point measurements without any relevance?

3. Aptamer alignment by Clustal: So, which is the proposed consensus sequence? Analysis softwares for aptamers are available, including MEME, which are much more suitable than Clustal is.

Reviewer #3: The authors replied to all my comments, the paper appears more straightforward and can be accepted for publication.

7. PLOS authors have the option to publish the peer review history of their article (what does this mean?). If published, this will include your full peer review and any attached files.

Reviewer #2: No

Reviewer #3: No

---

## [Author Response · Author response to Decision Letter 2]

7 Oct 2021

Response to reviewers comments: 

Reviewer #2: I agree that the authors made some progress, but I have yet serious reservations regarding the scientific quality of this manuscript.

Comment 1. Authors did not understand the point of my concerns of a large number of PCR cycles for reestablishing the X-DNA pool. The point is that over amplification, i.e. by 25 cycles, for reestablishing 100 % from 1 % results in a loss of pool and sequence heterogeneity and selection of artefacts. This is shown in the alignment analysis that authors now provided. It is expected that there are conserved consensus motifs within a part of the previous random region, but not that practically the entire random region makes part of the random regions. If this would be like that, how authors explain the differences in binding affinities.

Reply to comment 1: Due to valuable comment, we understand that there is a misleading for the selection of aptamers. We didn’t use SELEX for the aptamer selection, we used X-aptamer kit and perform according to manifacturer instructions. The SELEX refers to a different process which itself implies multiple rounds of selection process by using PCR. We did not use SELEX which is a round of PCR amplification. We performed a bead-based aptamer selection. For more information we kindly advise you to watch the process from the following link. (https://www.youtube.com/watch?v=CgfX-KmXdA0). Therefore, we decided to mention the process in the manuscript in a more appropriate way as “We performed a bead-based aptamer selection against two targets, bacterial /eukaryotic GHRH 1-44 and GHRH 1-29 peptide. In addition we replaced “first SELEX” with “first aptamer selection”, and replace “second SELEX” with “second aptamer selection”. The different cycle numbers of PCR were only to determine the proper cycle number for each fraction to send clean product to sequencing with no overamplification. 

Comment 2. Figure 3c: Where is the control (scrambled sequence aptamer) for these assays. Increase of an unspecific aptamer concentration will augment binding, however cannot be saturated. This needs to be shown. I understand that a non-linear curve using a zero point needs to be performed. Please, check for that TKY2.T1.T2, TKY.T1.01 and TKY.T1.02. The fittings are just wrong. Are these just one-point measurements without any relevance?

Reply to comment 2: According to valuable comment, we synthesized scramble sequence aptamer that has similar �G value. The sequence was given in the revised manuscript. We performed again dot-blotting with scramble sequence aptamer. Due to new dot-blot results, we analyze the Kd levels again and fit the results at GraphPad prism program. All the new results were given in the revised version of the manuscript. Although it is not a single measurement, we repeat dot-blot via using scramble aptamer. 

Comment 3. Aptamer alignment by Clustal: So, which is the proposed consensus sequence? Analysis softwares for aptamers are available, including MEME, which are much more suitable than Clustal is.

Reply to comment 3: In our previous revision, we performed Cluster alignment program to determine the consensus sequence. However, due to comment we performed MEME analysis for all aptamers instead of Clustal analysis. According to MEME analysis 5’ and 3’ constant sequences and random sequence was given in the revised version of the manuscript.

---

## [Decision Letter · Decision Letter 3]

4 Nov 2021

Synthesis and characterization of novel ssDNA X-Aptamers Targeting Growth Hormone Releasing Hormone (GHRH)

PONE-D-21-00439R3

Dear Dr. GURKAN,

We’re pleased to inform you that your manuscript has been judged scientifically suitable for publication and will be formally accepted for publication once it meets all outstanding technical requirements.

Kind regards,

Paulo Lee Ho, Ph.D.

Academic Editor

PLOS ONE

Additional Editor Comments (optional):

Reviewers' comments:

Reviewer's Responses to Questions

**Comments to the Author**

1. If the authors have adequately addressed your comments raised in a previous round of review and you feel that this manuscript is now acceptable for publication, you may indicate that here to bypass the “Comments to the Author” section, enter your conflict of interest statement in the “Confidential to Editor” section, and submit your "Accept" recommendation.

Reviewer #2: All comments have been addressed

2. Is the manuscript technically sound, and do the data support the conclusions?

Reviewer #2: Yes

3. Has the statistical analysis been performed appropriately and rigorously? 

Reviewer #2: Yes

4. Have the authors made all data underlying the findings in their manuscript fully available?

Reviewer #2: Yes

5. Is the manuscript presented in an intelligible fashion and written in standard English?

Reviewer #2: Yes

6. Review Comments to the Author

Reviewer #2: My comments were addresssed, and the paper is ready for publication. All criteria for publication in PloS ONE have been met.

7. PLOS authors have the option to publish the peer review history of their article (what does this mean?). If published, this will include your full peer review and any attached files.

Reviewer #2: No

---

## [Editor Report · Acceptance letter]

10 Jan 2022

PONE-D-21-00439R3 

Synthesis and characterization of novel ssDNA X-Aptamers Targeting Growth Hormone Releasing Hormone (GHRH) 

Dear Dr. Coker-Gurkan:

I'm pleased to inform you that your manuscript has been deemed suitable for publication in PLOS ONE. Congratulations! Your manuscript is now with our production department. 

Kind regards, 

on behalf of

Dr. Paulo Lee Ho 

Academic Editor

PLOS ONE